# Potential Benefits and Harms of Intermittent Energy Restriction and Intermittent Fasting Amongst Obese, Overweight and Normal Weight Subjects—A Narrative Review of Human and Animal Evidence

**DOI:** 10.3390/bs7010004

**Published:** 2017-01-19

**Authors:** Michelle Harvie, Anthony Howell

**Affiliations:** The Nightingale Centre, University Hospital of South Manchester NHS Foundation Trust, Southmoor Road, Manchester M23 9LT, UK; Tony.Howell@ics.manchester.ac.uk

**Keywords:** intermittent energy restriction, fasting, weight loss, weight gain

## Abstract

Intermittent energy restriction (IER) has become popular as a means of weight control amongst people who are overweight and obese, and is also undertaken by normal weight people hoping spells of marked energy restriction will optimise their health. This review summarises randomised comparisons of intermittent and isoenergetic continuous energy restriction for weight loss to manage overweight and obesity. It also summarises the potential beneficial or adverse effects of IER on body composition, adipose stores and metabolic effects from human studies, including studies amongst normal weight subjects and relevant animal experimentation. Six small short term (<6 month) studies amongst overweight or obese individuals indicate that intermittent energy restriction is equal to continuous restriction for weight loss, with one study reporting greater reductions in body fat, and two studies reporting greater reductions in HOMA insulin resistance in response to IER, with no obvious evidence of harm. Studies amongst normal weight subjects and different animal models highlight the potential beneficial and adverse effects of intermittent compared to continuous energy restriction on ectopic and visceral fat stores, adipocyte size, insulin resistance, and metabolic flexibility. The longer term benefits or harms of IER amongst people who are overweight or obese, and particularly amongst normal weight subjects, is not known and is a priority for further investigation.

## 1. Introduction

Excess energy intake, weight gain and subsequent adiposity are consistently linked to illness, disability and mortality [1,2,3]. Randomised trials demonstrate that intentional weight loss reduces type 2 diabetes [4], all-cause mortality [5] and increases cognitive [6] and physical function [7]. The health benefits of weight loss and energy restriction in these human clinical trials are supported by a century of laboratory research in rodents, which has established that energy restriction (ER) prevents age-related disease including tumours, cardiovascular disease, diabetes and dementia; retards aging-related functional decline; and increases lifespan [8].

Most human and animal studies on weight loss have involved continuous energy restriction (CER) administered on a daily basis. More recently, interest has focussed on intermittent energy restriction (IER) defined as periods of energy restriction interspersed with normal energy intake.

IER is of potential interest to manage obesity and its metabolic sequelae and also for normal weight subjects hoping to optimise their health independent of weight loss for two main reasons: firstly, IER only requires the individual to focus on ER for defined days during the week which is potentially more achievable than the standard approach of CER which is associated with poor compliance [9]; and, secondly, many beneficial metabolic effects achieved with weight loss and energy restriction are related to the energy restriction per se and are attenuated when the individual is no longer in negative energy balance [10,11]. It is therefore possible that repeated spells of marked ER for short spells during the week could provide metabolic benefits to post obese individuals beyond the period of weight loss who are no longer in negative energy balance. IER may also provide metabolic benefits for normal weight subjects, although this requires further investigation. 

The most studied IER approaches are either two consecutive days of ER per week (“two day”) [12,13] or alternate days of ER (ADER) [14], typically with a restriction which is 60%–70% below estimated requirements, or a total fast on alternate days [15,16,17,18]. Confusingly, all three regimes have been called “intermittent fasting” in the literature. In this review we will use the term intermittent energy restriction (IER) to cover all of these approaches, “two day” for two consecutive days per week, alternate day energy restriction (ADER) when restriction is 60%–70% every other day and intermittent fasting (IF) when there is no energy intake on alternate days. It is important to distinguish IF from other IER regimens which allow food on restricted days as IF may evoke greater metabolic fluctuations (for example increased free fatty acids (FFA) and ketones) [19,20], induce stress in individuals [21] and may be associated with hyperphagia during non-restricted days [19].

The heightened scientific and lay interest in IER amongst overweight and normal weight subjects [22,23], indicates a need to summarise and evaluate the effectiveness and metabolic effects of IER compared with CER and assess the safety of IER. Recent IER reviews highlight the relative paucity of human data and concluded that IER is comparable to CER for weight loss with little evidence of a metabolic advantage [24,25]. However in most studies IER and CER were not matched for energy intake. In addition, reviewers did not consider the effects of IER amongst normal weight subjects or the important issue of potential harm of IER. This narrative review will examine how IER compares with CER in terms of weight loss, metabolic changes and safety.

## 2. Methods

A Medline search was undertaken from 1946 to October 2016 using the search terms “intermittent” or “alternate day” or “modified” or “fasting” or “diet” or “calorie” or “energy restriction” linking with “body weight”, “body fat”, “hepatic fat”, “ectopic fat”, “fat free mass (FFM)”, “resting energy expenditure (REE)”, “insulin resistance”, “insulin sensitivity”, “metabolic flexibility” (Table 4). 

We include trials of IER with short periods of at least 50% energy restriction (≤7 days) interspersed with days of normal eating (≤10% energy restriction), but not studies of Ramadan, restriction for a few hours within the day (time restricted feeding) or studies with extended restricted periods, e.g., 2–5 weeks of dieting and not dieting which are testing different behavioural paradigms to weekly IER.

To compare adherence and weight loss success between IER and CER we include only randomised comparisons of IER and CER amongst free living individuals where the prescribed diets had been matched for overall energy intake. IER and IF are likely to have different effects on metabolic outcomes of interest, e.g., insulin resistance and REE during restricted and non–restricted phases. For this reason, the metabolic effects of IF and IER have only been reported from studies where authors have stated that measurements were undertaken on restricted or non-restricted days, which is critical to describe the overall metabolic effects of the IF and IER regimens.

## 3. Results

### 3.1. Is IER Associated with Greater Weight Control than CER?

#### 3.1.1. Weight Loss amongst People with Overweight or Obesity

We identified 13 randomised comparisons of IER and CER [12,13,14,18,26,27,28,29,30,31,32,33,34]. Seven of the studies were excluded because energy intake was not equivalent between the IER and CER groups [18,28,29,30,31,32,34]. ADER is the most studied IER amongst humans [25], however most of these studies summarised in recent reviews [24,25] have either a no treatment comparison group or no comparison group and so were not included.

We present data on adherence and weight change with IER vs. CER for the remaining six studies [12,13,14,26,27,33]. These trials tested different IER regimens; three tested a two day IER [12,13,33], one a four day IER [27], one an alternating pattern of three to seven days of IER per week [26] and one study tested ADER [14]. Most of the IER regimens advised healthy eating on the non-restricted days, with the exception of Carter et al. [33] (Table 1 and Table 2). The trials were relatively small with between 32 and 115 subjects randomised and between 25 and 88 completers within the trials. The trials were not powered to detect differences in body weight, and were relatively short duration (12–26 weeks). Drop out from the studies was between 0% and 40% and mainly comparable between the IER and CER groups, and to previous reports within CER studies [35]. Five of the studies reported an intention to treat analysis to account for these drop outs [12,13,14,27,33].

All of the selected studies demonstrate comparable reductions in body weight [12,14,26,27,33] between IER and CER. Four of the studies report equivalent reductions in body fat [12,26,27,33]; whilst one reported a greater loss of body fat with two different low carbohydrate IER regimens compared with CER over a four-month period [13]. The two IER regimens in this study allowed two consecutive days per week of either a low carbohydrate, low energy IER (70% ER, 2.7 MJ, 50 g carbohydrate/day) or a less restrictive low carbohydrate IER which allowed ad libitum protein and ad libitum monounsaturated fatty acids (MUFA, 55% ER, 4.78 MJ, 50 g carbohydrate/day). Both had 5 days of a healthy Mediterranean type diet, (45% energy from low glycaemic load carbohydrates, 30% fat; 15% MUFA, 8% polyunsaturated fatty acids and 7% saturated fatty acids). The IER regimens were compared to an isoenergetic 25% CER Mediterranean type diet. The differences in overall carbohydrate intake between the diet groups were modest (41% and 37% of energy for the two IER diets compared to 47% of energy for CER), which is unlikely to account for differences in adherence and reductions in adiposity between the diets [36].

#### 3.1.2. Adherence to IER and CER amongst People with Overweight or Obesity

Adherence to diets within trials is notoriously difficult to assess due to missing dietary records and well documented underreporting amongst overweight subjects [37]. This notwithstanding, dietary records in four of the studies [12,13,26,27] provided some information of the relative adherence and overall energy intake of IER vs. CER which was broadly comparable between the two groups.

Two of the studies of a two-day IER explored weekly adherence to the ER days and also intake on intervening non-restricted days to assess whether there is any evidence of energy compensation on these days [12,13]. The first study used a simple IER with two consecutive restricted days of 2.73 MJ, from milk, fruit and vegetables. An intention to treat analysis assuming women who left the study or who did not complete food records were non-adherent reported that mean (95% CI) 66% (55%–77%) of the potential IER days were completed. The low carbohydrate IER tested in the second study [13], allowed a larger range of foods than the previously tested regimen and appeared to have a greater adherence; mean (95% CI) potential IER days completed for the low carbohydrate, low energy IER and the less restrictive IER were respectively 76% (67%–81%) and 74% (64%–84%).

Neither of the IER tested were associated with compensatory hyperphagia on the non-dieting days. These trials have instead reported an important “carry over effect” of reduced energy intake by ~20% on non-restricted days (Figure 1). Energy intake on the non-restricted days of the IER regimen was similar to the planned 25% restriction of the CER regimen [12,13]. Food records amongst the CER group in this study showed that 55% were achieving their daily 25% and an overall 25% CER. Thus the greater loss of fat reported with the two day low carbohydrate IER diets compared to CER in the 2013 study appears to be linked to better dietary adherence with IER vs. CER, partly linked to good adherence to the two restricted days each week and the spontaneous restriction of energy intake on non-restricted days [13].

Energy intake was not presented in the RCT of ADER compared to CER [14], however previous reports of ADER have similarly found a small carry over effect with a 5% reduction in energy on the non-restricted days of the regimen [38]. A number of studies were testing combined IER/CER and exercise interventions [13,26,33] and reported exercise to be feasible alongside both IER and CER diets, but there are no reported interactions between diet allocation and levels of activity and weight loss.

#### 3.1.3. Maintaining Weight Loss amongst People with Overweight or Obesity

Dietary approaches must demonstrate the ability to maintain weight loss. Estimates of successful weight loss maintenance with CER (defined as >10% weight loss maintained at ≥12 months), vary between 20% and 50% depending on the level of support provided at later time points [9,39]. There are few data on weight loss maintenance using IER. One month of a weight maintenance IER (one day of ER and six days of an ad libitum Mediterranean type diet per week) successfully maintained reductions in weight and of insulin resistance which had been achieved with three months of a weight loss IER (two days of ER and five days of an ad libitum Mediterranean type diet) in the 2013 IER trial described above [13]. However, one month is too short a period to draw any conclusions about the longer term efficacy of IER. They had previously reported that after six months of dieting, 58% of the IER group compared to 85% of CER subjects planned to continue the diet allocated at randomisation [12]. Unpublished data from this trial show, by 12 months, 33% of the initial IER randomisation group were still undertaking one or two days of IER each week. An intention to treat analysis based on last observation carried forward showed no difference in the percentage of women in the IER and CER groups losing 5%–10% of body weight (25% vs. 30%) or losing 10% or more at 12 months (19% vs. 19%, χ^2^ = 0.370, *p* = 0.83) (Harvie et al. unpublished).

In summary, randomised trials to date have found IER to be equivalent to CER for weight loss in six studies [12,13,14,26,27,33] and superior to CER for reducing body fat in one study [13]. All studies were relatively small and in the author’s estimations only powered to show absolute differences of 4%–5% between the groups. Larger studies would be required to investigate if there are smaller differences in weight loss between IER and CER regimens (2%–3% weight loss differences), which would still be clinically meaningful. Studies were mainly conducted amongst women and included motivated groups of subjects who had answered adverts [14,26,27], or were recruited from a clinic for women at increased risk of breast cancer [12,13]. Interventions have been short term at ≤26 weeks, and tested under highly supervised conditions with high levels of dietetic support and sometimes meal provision [14,27]. Larger, longer term, real world weight loss trials are required to inform how IER performs longer term in a variety of settings in comparison to the standard approach of CER.

### 3.2. Prevention of Weight Gain amongst Normal Weight Subjects

There are currently no randomised data to compare the relative efficacy of IER vs. CER to prevent adult weight gain amongst normal weight subjects. Two studies have tested the metabolic effects of IER amongst groups of normal and overweight subjects (BMI 20–30 Kg/m^2^). These studies have been designed to assess short term metabolic effects, mainly for two to three weeks and involved highly controlled situations in which participants were instructed to increase dietary intake on non–restricted days to ensure they did not have an overall energy deficit [15,40]. The studies report sustained hunger with IF [15,40], and difficulties maintaining daily living activities during restricted days of an ADER regimen [41]. This suggests limited compliance and efficacy of these specific regimens amongst cohorts of normal and overweight subjects. However, other patterns of IER, e.g., one restricted day per week, may be better tolerated and warrant study. 

## 4. Metabolic Effects of IER vs. CER

### 4.1. Adipose Stores and Adipocyte Size

#### 4.1.1. Human Studies

Reductions in adiposity, specifically visceral and ectopic (i.e., hepatic/abdominal and intramuscular) fat stores are a therapeutic target of ER. Hepatic and visceral fat stores rapidly mobilise with marked ER as they are thought to be more sensitive to the lipolytic effects of catecholamines during negative energy balance than subcutaneous fat [41]. Marked CER (>50% ER) is related to rapid decreases in hepatic fat in people with obesity [42,43]. Lim et al. [43] reported a 30% reduction in hepatic fat after seven days of a 60%–70% CER in subjects with type 2 diabetes (nine men and two women), which normalised hepatic insulin sensitivity. There are currently no human data concerning the long term or chronic effects of IER on hepatic, intra-abdominal and intramyocellular triglyceride stores. Reports of significant reductions in hepatic fat stores (−29%) after two days of ER and carbohydrate restriction in men and women with obesity [42], suggest reductions could occur during the repeated spells of restriction with IF/IER each week. Such reductions may account for the reported improvements in homeostatic model assessment (HOMA) insulin resistance, i.e., hepatic insulin resistance with IER described below [12,13], (see Section 4.4) but require further study. 

In contrast, short term fasting studies raise the possibility of harm with IER amongst normal weight subjects. Periods of IER each week will induce lipolysis and fluxes in FFAs. The 1–2 days of IF/IER each week will lead to large fluxes in FFA which are typically three-fold greater than those seen after a normal overnight fast [44], and will be larger with IF rather than IER [19]. These fluxes can lead to skeletal muscle insulin resistance. Single bouts of total fasts (24–48 h) in non-obese subjects have been associated with modest increases in hepatic and intramyocellular triglyceride content which are not seen after the normal 12 h overnight fast. Specifically a single spell of fasting of between 24–48 h leads to modest increases in intramyocellular triglycerides (2.4%–3.6%) but not hepatic fat in non-obese premenopausal women, mainly in the second 24 h period of fasting [45], whilst men have modest increases in hepatic fat (0.42%–0.74%) within the first 24 h of fasting, but do not accumulate intramyocellular triglycerides [45,46]. The clinical significance of the modest changes in hepatic triglycerides in men [46], and intramyocellular triglycerides in women [45] in these short term studies is not known. Some [45,47] but not all [48] studies have associated increased intramyocellular triglycerides with reduced insulin sensitivity upon refeeding amongst women. Possible mechanisms for increased hepatic fat with fasting in men include reduced apolipoprotein B-100 production and hepatic lipid export, and/or impaired mitochondrial function and fat oxidation resulting from increased oxidative stress, with diversion of fatty acids for esterification [49]. The effects of repeated IER each week on hepatic and intramyocellular triglyceride stores and whole body insulin sensitivity needs to be assessed in longer term studies and also amongst people who are overweight or obese.

#### 4.1.2. Animal Studies

Studies in rodents report mixed effects of IER vs. CER on hepatic and visceral fat stores. One month of alternate days of either fasting or a 75% or 85% energy restriction without an overall energy restriction in female C57BL/6J mice did not change weight or total amounts of body fat, but led to redistribution of fat from visceral (−40%) to subcutaneous stores (+65%) [50]. A similar investigation amongst male C57BL/6J mice did not find that alternate days of fasting or ER had effects on total or visceral fat stores. However, in this study, alternate days of a 50% ER and ad libitum feeding reduced fat cell size in the inguinal (subcutaneous) fat pads by 50% and in epididymal (visceral) fat pads by 35% [51], despite there being no overall energy restriction. Marked reductions in fat cell size are thought to reduce risk of inflammation and metabolic diseases [52].

However in other animal models (four week old male Sprague Dawley rats and LDL-receptor knockout mice) IF regimens reduced energy intake and weight but increased visceral fat, fat cell size and had adverse effects on insulin sensitivity compared to heavier ad libitum fed animals [53,54]. The variable effects of IER vs. CER on fat stores in different animal models means extrapolating findings from specific animal models to the human situation is problematic. The adverse effects of IF in these studies may be because IF animals adopt a gorging pattern of eating which in turn can shift normal night time grazing to a pattern of overfeeding during daylight hours. This disturbance of circadian rhythms may lead to the reported accumulation of abdominal and hepatic fat and adverse metabolic effects [55]. The adverse effects of fasting and ER seen in these rodent studies are important to consider, but may not be an issue for humans. In contrast to rodent studies, people who are overweight or obese undertaking IER appear to reduce intake on the non-restricted days [12,13,38] and do not display compensatory overfeeding (see Section 3.1). The effects of IER and CER on circadian rhythm is important, however this has not been studied.

### 4.2. Fat Free Mass

#### Human Studies

Weight loss and weight maintenance diets should reduce body fat stores and, as far as possible, preserve FFM to maintain physical function and attenuate declines in resting REE and help to prevent weight gain. CER is known to reduce FFM in addition to body fat. Typically 10%–60% of weight reduction using CER is FFM, depending on initial body fat, the degree of energy restriction, extent of exercise and protein intake [56]. Proponents of IER and IF diets claim they may preserve FFM more than CER from cross study comparisons of IER and CER interventions which may have allowed our Palaeolithic hunter gatherer ancestors to survive spells of food shortage [57]. However the concern is that spells of severe restriction with IER and IF (i.e., fasting or intakes of <2.0 MJ/day) could lead to greater losses of FFM than the modest daily energy restriction with CER. There are, however, few data to inform this question, as the modest sized IER trials undertaken are unlikely to be powered sufficiently to demonstrate difference in FFM loss [58]. Weight loss trials amongst people who are overweight or obese suggest losses of FFM with IER and CER are equivalent within the bounds of small numbers and are dependent on the overall protein content of the IER and CER diet rather than the pattern of energy restriction [59]. The first IER trial reported an equivalent loss of weight as FFM between IER and CER (both 20% of weight loss) when both diets provided 0.9 g protein/kg body weight [12]. Likewise the 2013 trial reported equal losses of FFM (both 30% of weight loss) with a standard protein IER (1.0 g protein/kg body weight) compared to a standard protein CER (1.0 g protein/kg body weight) [13]. There was however a greater preservation of FFM (20% of weight loss) with a high protein IER (1.2 g protein/kg weight) compared to the standard protein CER (30% with 1.0 g protein/kg body weight (*p* < 0.05) [13]. Likewise Hill et al. [26] reported 27% of weight loss from FFM with IER and CER which both provided 0.7 g protein/kg body weight. Studies of ADER reported the proportion of weight lost as FFM as low as 10% in women with obesity [60] and as high as 30% amongst non-obese subjects [61]. Subsequent studies show that exercise helps to retain FFM amongst subjects undergoing IER [26] and ADER [62] which is well documented with CER [63].

One study assessed muscle protein turnover before and after 14 days of alternate day fasting in normal weight healthy men [17]. This study reported lowered mechanistic target of rapamycin (mTOR) phosphorylation in muscle which was thought to reflect decreased muscle protein synthesis and a failure to reduce muscle proteolysis. These changes could lead to a reduced muscle mass, thus suggesting that long term alternate day fasting could lead to reduced muscle mass in normal weight subjects [17]. 

### 4.3. Resting Energy Expenditure

#### Human Studies

Resting energy expenditure accounts for 60% to 75% of the total daily energy requirement in an individual, thus it is important in determining overall energy balance and whether an individual is weight stable or gaining or losing weight. REE is known to be reduced during CER in association with reduced FFM and fat mass [64], as well as to reduced circulating leptin and thyroid hormones and sympathetic nerve activity [65]. Total energy expenditure may be reduced 10% within two-weeks of starting 25% CER [65].

There are few data on the effects of IER on REE. REE could be acutely decreased during the short restricted periods each week, which could normalise during the normal eating days of the week. However, an increase in REE of ~5% is seen during the first days of starvation [66], perhaps as a result to the increased energy cost of fatty acid recycling, glucose storage, gluconeoegenesis, increased sympathetic nervous activity and catecholamine concentrations [67]. Studies to date have assessed REE after non-restricted days of IER and have mostly shown reductions in REE amongst subjects who are overweight or obese [26,28,68] and overweight or a normal weight [15,17]. One exception is a recent trial of ADF amongst 26 subjects with obesity, where REE decreased with CER but not IER [18]. Most studies suggest IER evokes the same adaptive response as CER at least on non-restricted days. Future studies should assess the effects of IER on REE during restricted days to assess the overall impact of IER on metabolic rate. 

### 4.4. Peripheral and Hepatic Insulin Resistance

#### 4.4.1. Human Studies

Insulin acts on skeletal muscle to increase glucose uptake and inhibit protein catabolism, on adipose tissue to increase glucose uptake, lipogenesis, lipoprotein lipase and uptake of triglycerides, and on the liver to reduce lipolysis, gluconeogenesis and increases glycogen synthesis. Obesity is associated with both peripheral and hepatic insulin resistance where normal or elevated insulin levels have an attenuated biological response in these tissues [69]. Studies in obese, overweight and normal weight subjects have assessed the effects of IER on whole body, peripheral and hepatic insulin sensitivity using a variety of methods, with variable results. 

We assessed HOMA insulin resistance, a measure of hepatic insulin sensitivity, in two RCTS of a two-day IER versus CER amongst subjects who are overweight or obese. As indicated above, the first trial compared IER (two consecutive days of 70% ER per week) to an isoenergetic CER (25% ER Mediterranean type diet seven days per week) amongst 105 healthy women [12]. The IER led to greater reductions in HOMA compared to CER when measured on the morning after five normal eating days. The mean (95% CI) % change in HOMA over six months IER was −24 (−35 to −13)% and CER was −4 (−20 to +12)%, (*p* = 0.001). We also measured HOMA on the morning after the two energy restricted days, which showed an additional 25% reduction compared with CER at this time. These differences in insulin sensitivity occurred despite comparable reductions in body fat between the groups (IER −4.5 vs. CER −3.6 kg, *p* = 0.34). 

The follow up study tested two low carbohydrate IER regimens which allowed two consecutive days per week of either a low carbohydrate, low energy IER (70% ER, 2.7 MJ, 50 g carbohydrate per day) or a less restrictive low carbohydrate IER which allowed ad libitum protein and MUFA (55% ER, 4.18 MJ, 50 g carbohydrate per day). These regimens led to equivalent reductions in body fat which were both greater than CER as described above (see Section 3.1) [13]. However, reductions in serum insulin and HOMA insulin resistance measured after non-restricted days were significantly lower than CER only with the lower energy IER (*p* = 0.02), but not the less restrictive IER regimen (*p* = 0.21). The reasons for the apparent greater improvement in insulin resistance with more restrictive IER is independent of changes in body fat and may be specifically linked to the more marked energy restriction on restricted days (70% vs. 55%). A recent small trial amongst 26 subjects with obesity reported successful weight loss with IF, −8.8 (0.9)% or a 16% CER, −6.2 (0.9)%. However, neither group experienced changes in insulin resistance assessed using an insulin-augmented frequently sampled intravenous glucose tolerance test measured after a non-restricted day [18]. 

There are few data of the effects of IER vs. CER on glucose control amongst overweight and obese individuals with type 2 diabetes Ash et al. (2003) reported that a four day IER over 12 weeks led to equivalent reductions in percentage body fat (see Section 3.1) and in HbA1c compared with isoenergetic CER although this small study may have been underpowered to show significant differences [27]. Carter reported equivalent reductions in HbA1c in individuals with Type 2 diabetes with 12 weeks of IER or CER which had been achieved with greater (albeit non-significant) reductions in insulin medications within the IER group [33]. Williams et al. [57] assessed the effect of enhancing a standard 25% CER diet with periods of 75% ER (either five days per week every five weeks or one day per week for 15 weeks). Predictably, additional periods of ER increased weight loss. The five days per week intervention resulted in the greatest normalisation of HbA1c, independent of weight loss, suggesting a potential specific insulin-sensitising effect of this pattern of IER added to CER [30]

Three studies assessed the effects of two to three weeks of IF (alternating 20–24 h periods of a total fast interspersed with 24–28 h periods of hyperphagia (175%–200% of estimated energy requirements) [16,17,18]. They were designed to ensure there was no overall energy deficit or weight loss, and results have varied between the studies. Halberg et al. [16] assessed the effects of two weeks of IF (a total fast for 20 h from 22.00 and ending at 18.00 the following day) in eight overweight young men. Improvements in insulin mediated whole body glucose uptake and insulin induced inhibition of adipose tissue lipolysis assessed using a euglycaemic hyperinsulinaemic clamp were seen when measured after two normal feeding days, which the authors suggested may be related to higher adiponectin concentrations seen during the 20 h fast [16]. Soeters et al. [17] tested an identical two week IF intervention in normal weight men in a cross over design. However IF was not associated with changes in peripheral glucose uptake or hepatic insulin sensitivity assessed with a hyperinsulinaemic clamp, or lipid or protein metabolism. Heilbronn et al. [47] assessed the effects of three weeks of IF (alternating 24 h total fast and 24 h ad libitum feeding) amongst 16 overweight men and women. Glucose uptake during a test meal was assessed at baseline after a 12 h fast and after three weeks of IF on the morning after a fasting day, i.e., after a 36-h fast. Men had a significant reduction in insulin response and improved glucose uptake and insulin sensitivity, whilst women had impaired glucose uptake and apparent skeletal muscle insulin resistance. This observation is likely to be related to greater fluxes of FFA during this 36-hour fast amongst fasting women [70], which most likely reflects a normal physiological adaptation to fasting rather than a cause for concern. Reduced glucose uptake in skeletal muscle limits the competition between skeletal muscle and the central nervous system and other glucose obligate tissues for circulating glucose in situations with low glucose availability, which reduces gluconeogenesis and has protein sparing effects in turn has protein sparing effects [71]. The short term studies of IF outlined above report mixed results on peripheral and hepatic insulin sensitivity and raise the possibility of different responses to IF according to gender. Further studies are required using robust measures of insulin sensitivity. 

#### 4.4.2. Animal Studies

Variable effects of IF regimens on insulin sensitivity have also been reported in animal studies [72,73,74]. Higashida et al. [74] tested whether energy restriction with IF could prevent the development of muscle insulin resistance induced by a high fat diet. Young male Wistar rats were given a high fat diet for four weeks and then allocated to a continued high fat diet (*n* = 12) or an IF with alternate days of fasting and an ad libitum high fat diet for six weeks (*n* = 12). These animals were compared with a group who had been fed a 36% CER chow diet for the 10 week study (*n* = 12). The IF and CER rats had a reduced weight (IF −27%, CER −14%), and reduced intra-abdominal body fat (IF −39%, CER −50%) compared to the high fat diet fed animals. IF failed to improve insulin stimulated glucose uptake in muscles (measured after a feed day) despite their lower adiposity. Both IF and ad libitum fed animals had reductions in muscle GLUT–4 proteins compared to CER (−30% and −42%). However IF animals had increased serum concentrations of adiponectin (+92%) and reduced HOMA insulin resistance (−49%) compared to the high fat fed animals indicating improved hepatic insulin sensitivity with IF [74]. Thus, in this animal model, IF had a favourable effect on hepatic, but not muscle insulin resistance compared with CER.

The apparent greater reductions in HOMA insulin resistance with a two day IER compared with CER in premenopausal women who are overweight or obese [12,13], and in some relevant animal models [74] raises the possibility that IER may improve hepatic insulin sensitivity. However, IER did not appear to evoke greater improvements in insulin sensitivity than CER in three other human comparator trials [18,27,33].

IF has been shown to have variable effects on peripheral and hepatic insulin stimulated uptake of glucose in non-obese subjects. The health implications of repeated short term increases in FFA and increases in peripheral insulin resistance with IF and IER each week are not known and need further investigation. This may be particularly important for groups which experience the largest fluxes of FFA, i.e., normal weight individuals and women [75].

## 5. Metabolic Flexibility

Periods of energy restriction or prolonged exercise switch liver, skeletal muscle and cardiac tissues to fat oxidation, and the catabolism of amino acids, whilst the post prandial state favours glucose uptake and oxidation. The reciprocal regulation of fat and glucose oxidation is controlled systematically by insulin and glucagon, and in response to changes in cellular levels of metabolites such as fatty acids, pyruvate, citrate and malonyl CoA which regulate mitochondrial enzymes. Energy metabolism is considered to be optimal when the body can readily switch between oxidising glucose or fat in response to nutrient availability and physiological stress [76]. This is considered to maintain metabolic health and the optimal cellular functioning. The switch in energy metabolism is known as metabolic flexibility. Metabolic inflexibility is seen in overfed individuals who do not easily switch between fat and glucose oxidation. There is simultaneous oxidation of fat, glucose and amino acids all of which increase oxidative stress, diacylglycerols, ceramides, and acylation of mitochondrial proteins, which in turn results in perturbations of mitochondrial function. Metabolic inflexibility is thought to be the root cause of insulin resistance [76].

Distinct periods of ER interspersed with normal energy intake each week may be akin to hunter-gatherer lifestyles and may promote maintenance of metabolic flexibility compared to standard daily diets, especially since IER contains longer periods of ER than our usual overnight fast. A recent study in rats of ADF supports this notion. Male Wistar rats were subjected to 48 days of IF (eight repeated cycles of three days of fasting and three days refeeding) or an isoenergetic 20% CER. The IF mice showed up regulation of genes for both lipid storage (PPARγ 2 and *Fsp27*) and fat oxidation (MCPT1) reflecting good metabolic flexibility with increased fat oxidation during fasting days and lipogenesis on non-restricted days of IF. These changes were not induced with an isoenergetic 20% CER [77].

In humans, six months of a 25% CER has been shown to improve metabolic flexibility, as evidenced by increased shift in fasting-to-postprandial concentrations of acyl carnitine (important for transfer of fatty acids into the mitochondrion prior to oxidation) [78]. There are currently no data of the effects of IER on metabolic flexibility in humans. 

### 5.1. Is IER Safe?

There are theoretical concerns that IER could promote erratic eating patterns, binging, and low mood. A recent systematic review of 15 clinical trials concluded that marked CER (>60%) reduced binge eating behaviour amongst overweight or obese individuals with pre-treatment binge eating disorder, and did not appear to trigger binge eating in those without previous binge eating disorder [79]. Hoddy et al. [68] reported reductions in depression and binge eating and improved body image perception after eight weeks of following ADER. In contrast, four weeks of IER (four consecutive days of a 70% ER and three days of ad libitum eating) amongst nine normal weight young women, classified as unrestrained eaters, resulted in increased feelings of hunger, worse mood, heightened irritability, difficulties concentrating, increased fatigue, eating-related thoughts, fear of loss of control and over eating during non-restricted days [80]. We have reported comparable reductions in profile of mood state scores for tension, depression, anger, fatigue and confusion, an increase in vigour and an overall decline in total mood disturbance with a two day IER and CER [12,13]. Thus existing data show IER can improve eating behaviours and mood amongst subjects with overweight and obesity, but may have the potential for harm amongst normal weight individuals with unrestrained eating styles.

Another frequent concern is whether the spells of marked energy restriction with IER could perturb the hypothalamic-pituitary-gonadal axis in women and alter the frequency and length of menstrual cycles. Such effects are likely to be related to the starting weight of the individual, overall energy balance and the number of consecutive restricted days with IER. The 2011 IER study amongst obese and overweight women reported a longer average menstrual cycle length in women following IER for six months (two consecutive days of 70% ER per week) compared to 25% CER group (29.7 ± 3.8 days vs. 27.4 ± 2.7 days, *p* < 0.005) [12]. A study amongst normal weight, sedentary, normal cycling women found that three consecutive days of a total fast during the mid-follicular phase affected luteinising hormone dynamics, but were insufficient to perturb follicle development, or menstrual cycle length [81]. The effect of IER on the reproductive axis amongst obese, overweight and normal weight subjects requires further study, especially regimens which include longer periods of energy restriction.

IER does not appear to limit an individuals’ ability to exercise. A 12 week combined ADER and exercise trial amongst subjects with obesity reported equal attendance to a supervised exercise programme (40 min of 75% max heart rate on three days per week) on both restricted and non-restricted days of ADER [62]. Similarly Carter et al. [33] reported a comparable increase in daily average step count in the IER and CER groups and Hill et al. [26] reported comparable and good adherence to a moderate intensity walking programme (five 20–50 min sessions of brisk walking 60%–70% max heart rate per week) amongst dieters undertaking IER and CER [26].

The majority of studied IER regimens have recommended healthy eating and not feasting on non-restricted days. Feasting on non-restricted days of IER may have adverse effects on health, despite weight loss. For example, a high fat ADER (45% fat) produced equivalent weight loss to a low fat ADER; 5.4 (1.5) kg vs. −4.2 (0.6) kg [82]. However, despite weight loss in this study, the high fat ADER group had decreased brachial artery flow mediated dilation which could increase risk of atherosclerosis and hypertension. 

Thus, limited data to date suggest that IER is not associated with disordered or binge eating, perturbation of the hypothalamic-pituitary-gonadal axis, and does not limit the ability to exercise amongst in individuals who are overweight or obese. However the safety longer term and amongst normal weight individuals is not known. 

### 5.2. Is There an Optimal IER Regimen?

The optimal duration, frequency and severity of ER needs to strike a pragmatic balance of being achievable, whilst also delivering supposed beneficial metabolic effects. There are numerous potential permutations of IER and IF which could be studied. IER is likely to be preferable to IF regimens amongst humans, as it is likely to have greater compliance and lower stress and cortisol responses [21]. IER regimens may need to provide some energy and protein intake on restricted days (i.e., 2.5 MJ and 50 g protein) to maintain nitrogen balance and FFM, which does not seem to be achieved with spells of total fasting [83]. IER evokes smaller fluctuations in FFAs and ketones than IF [19,20]. The latter is linked to short term impaired glucose tolerance during the resumption of normal feeding. The longer term implications of short term impairments in glucose tolerance with repeated IF each week is not known. 

The timing of energy intake during the restricted days of IER does not appear to be important for compliance and weight loss. Hoddy et al. [84] reported equal reductions in weight with a 75% ADER with either one meal at lunch or dinner or three small meals throughout the day. 

## 6. Conclusions

This review highlights a lack of high quality data to inform adherence and benefits or harms of IER vs. CER. Research findings and gaps in the evidence comparing IER to CER for weight control and metabolic health are summarised in Table 3. The few randomised comparisons of IER vs. CER amongst overweight and obese subjects report equivalent weight loss, with one trial of a two day low carbohydrate IER reporting greater reductions in body fat compared to CER [13]. These studies were not powered to detect a difference in loss of weight or fat, thus study finding are suggestive but not conclusive of no difference between IER and CER. No studies to date have tested whether IER can prevent weight gain amongst normal weight subjects, however IER regimens based on alternate days of total fasting or marked energy restriction (70% restriction) have not been well tolerated amongst normal and overweight populations (BMI 20–30 kg/m^2^) [15,40]. 

This review highlights numerous gaps in knowledge of the effects of IER vs. CER on ectopic and visceral fat stores, adipocyte size, FFM, insulin resistance, REE and metabolic flexibility, particularly amongst normal weight subjects. In the absence of these data, we have drawn on findings of short term studies and highlighted some potential beneficial or adverse effects. The variable and sometimes adverse effects of IER on fat stores and metabolism in some rodent models reported in this review are a concern. However, this may relate in part to shifts in night and day eating patterns and circadian rhythm [55], which may not translate to the human situation.

Future IER research requires two types of randomised comparison trials. Firstly, longer term RCTs of IER and CER (>6 months) to show whether IER is sustainable long term and has long term benefits or yet undiscovered harmful effects on weight, body composition, and metabolic health compared to CER. Secondly, detailed metabolic proof of principle studies in controlled conditions to assess the effects of IER and matched isoenergetic CER on FFM, hepatic and intramuscular fat stores, insulin sensitivity and metabolic flexibility using robust methodology such as DXA, MRI and insulin clamps. These studies need to assess the metabolic effects of IER during restricted and feed phases of the diet to fully characterise its biological effects amongst people of any weight. 

Well documented differences in metabolic responses to periods of fasting and marked energy restriction between pre-menopausal women (i.e., increased ketones and free fatty acids) compared to men and post-menopausal women suggest possible different metabolic responses, and perhaps better tolerance to IER within certain populations [85]. Future IER studies should include males, older subjects, individuals with morbid obesity or type 2 diabetes, as well as normal weight subjects. There is also a need to explore optimum patterns of restriction, e.g., two days per week, alternate days, five days per month [86] or other permutations and the best mode of restriction on the restricted and intervening days (e.g., low carbohydrate, low protein).

The popularity of IER within the general public coupled with the gaps in the evidence we have identified indicate that IER deserves further rigorous study. We do not know conclusively whether long term IER is a safe effective method of weight control for subjects who are overweight or obese or whether IER may confer health benefits to people of any weight independent of weight loss. High quality research comparing long term outcomes of IER and CER are required to ascertain any true benefits or detrimental effects which IER may have for controlling weight and improving metabolic health in the population.

## Figures and Tables

**Figure 1 behavsci-07-00004-f001:**
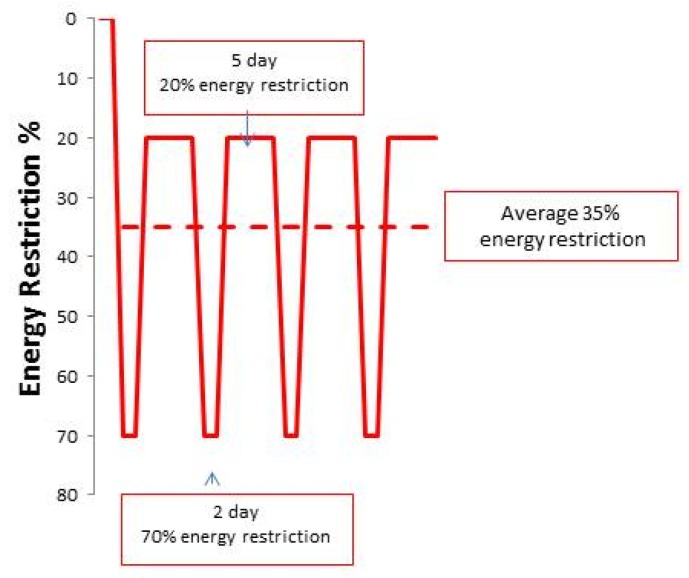
Degree of dietary energy restriction with IER in Manchester studies. The IER cohort undertook a 70% energy restriction on two consecutive days per week and additionally undertook an unplanned carry-over energy restriction to an average of 20% below their baseline intake on the remaining five days of the week (*solid line*). The restricted days and the unplanned carry-over energy restriction resulted in an equivalent overall 35% energy restriction over the trial period (*dashed line*).

**Table 1 behavsci-07-00004-t001:** Randomised weight loss trials of intermittent energy restriction compared to isoenergetic continuous energy restriction in people who are obese or overweight.

Reference	IER Regimens	Study Population		Study Design			Primary End Point and Power of the Study
		N, Gender, Age	Baseline BMI (kg/m^2^) Mean (SD) Range	Diet Groups (N)	Level of Support	Duration of Study	
Hill et al. 1989 [26]	3–7 day periods of alternating 70%, 60%, 45%, and 10% ER. Overall 40% CER	40 women (32 completers)	31 (3.0)	IER = 10 IER + exercise = 10 40% CER = 10 40% CER + exercise = 10	12 weekly group meetings. Menus provided.	12 weeks intervention and follow up 6 months after the weight loss programme	Weight loss No power calculation
Ash et al. 2003 [27]	4 consecutive days/week 50% ER (4.18 MJ liquid very low calorie diet 3 days/week, *ad lib* healthy eating) Overall 30% CER	51 men with Type 2 diabetes Age < 70 years	31.2 (3.4) 25–39.9	IER vs. 2 types of 30% CER IER = 14 30% CER = 20 (self-selected meals) 30% CER = 17 (pre-portioned meals)	Face to face visit the clinic dietitian and physician fortnightly and telephone contact with the dietitian on intervening weeks.	12 weeks	Weight loss and glycaemic control No power calculation
Varady et al. 2011 [14]	Alternate days of 75% ER (1.67–2.50 MJ/day) and AL low fat/American Heart Association diet: 30% kcal fat, 15% kcal protein, 55% kcal carbohydrate. Overall 25% CER	51 women and 9 men Age 35–65 years	32 (2.0) 25–39.9	IER = 15 (pre-portioned meals on fast days) 25% CER = 15 (pre-portioned meals) Exercise only =15 (180 minutes 70% max heart rate) Control = 15	No information	12 weeks	LDL and HDL particle size No power calculation
Harvie et al. 2011 [12]	2 consecutive days/week 70% ER (2.73 MJ/day, 50 g protein: 2 pints of milk, 1 portion of fruit and 4 portions of vegetables) 5 days ad libitum healthy eating. Overall 25% CER	107 premenopausal women Age 30–45 years	30.6 (5.1) 24–40	IER = 53 25% CER = 54	Fortnightly motivational phone calls and monthly clinical appointments with dietitian. Advised to maintain current levels of physical activity.	26 weeks	Insulin resistance 80% power to detect a 25% difference
Harvie et al. 2013 [13]	IECR: 2 consecutive days/week 70% ER (energy and carbohydrate restriction: 2.73 MJ/day, 70 g protein, 50 g carbohydrate) Overall 25% CER 2 days 60% ER. or IECR + PF: 2 consecutive days/week 50% ER (energy and carbohydrate restriction: ~4.18 MJ/day, 80 g protein, 50 g carbohydrate)	115 women aged 20–69 years	31 (5.0) 24–45	IECR = 37 IECR + PF = 38 25% CER = 40 2 days/week IECR vs. IECR + PF 1 days/week vs. isoenergetic CER diet	Fortnightly motivational phone calls and monthly clinical appointments with dietitian. Advised to achieve 5 × 45 min of moderate intensity physical activity per week—but achieved minimal changes in the three groups	13 weeks weight loss phase. 4 weeks weight maintenance phase.	Insulin resistance 80% power to detect a 20% difference
Carter et al. 2016 [33]	IECR 2 days per week 1.67–2.5 MJ/day (70%–85% restriction) and habitual eating for 5 days	63 30 men 33 women Type 2 diabetes Age > 18 Mean (SD) age 61 (7.5)	35 (4.8)	IECR = 31 + exercise (2000 steps) CER = 32 5.0–6.5 MJ (35%–45%) + exercise (2000 steps)	Asked to record dietary intake throughout the 12-week study. Fortnightly appointments with dietitian	12 weeks	HbA1c No power calculation

Mean (SD); IER, intermittent energy restriction; CER, continuous energy restriction; IECR, intermittent energy and carbohydrate restriction; IECR + PF, intermittent energy and carbohydrate restriction and ad libitum protein and MUFA.

**Table 2 behavsci-07-00004-t002:** Adherence, weight loss and changes in metabolic markers in randomised trials of intermittent energy restriction compared to isoenergetic continuous energy restriction in people who are obese or overweight.

				Outcomes		
Reference	Dropout % of Subjects	Dietary Adherence Methodology	Final Analysis	Weight change Mean (SD) % Weight Loss	Change in Body Fat and fat Free Mass (FFM) Method of Assessment	Metabolic Effects
Hill et al. 1989 [26]	12 weeks/6 months after intervention IER = 40%/60% IER + exercise = 0%/0% 40% CER = 20%/70% 40% CER + exercise = 20%/40% Combined IER + IER + exercise = 20%/30% Combined CER + CER + exercise = 20%/55%	12 week diet records Average daily intake IER = 4.97 (0.59) MJ IER + exercise = 4.58 (2.92) MJ 40% CER = 5.46 (1.49) MJ 40% CER + exercise = 4.59 (0.30) MJ *p* > 0.05	Completers analysis	12 week data Combined IER/CER diet only groups Weight −6.5 (0.9) kg (−7.6%) Combined IER/CER + exercise −8.6 (0.9) kg (−8.8%)	Body density from underwater weighing IER = CER Fat loss (kg): IER 6.0 (0.8) CER 6.1 (0.6) *p* > 0.05 Loss of FFM: Combined all groups 47.6 (1.1) to 46.0 (1.0) *p* < 0.05	Equal reductions in blood pressure and triglycerides with IER vs. CER. No change in insulin with IER or CER (*p* > 0.05) Reduced total cholesterol IER −14% vs. CER −6% (*p* < 0.05)
Ash et al. 2003 [27]	IER = 0% 30% CER (self-selected meals) = 0% 30% CER (pre-portioned meals) = 0%	24 hour recalls All groups mean (SD) reduction in average daily energy intake −2.36 (2.78) MJ ~30% energy restriction	Completers as no drop outs	IER = CER Combined IER/CER group 6.5 (6%)	DEXA IER = CER % Body fat loss: IER −2.0 (1.1)% CER (self-selected meals) −0.9 (1.4)% CER (pre-portioned meals) −2.6 (1.6) % (*p* = 0.41) FFM: no data	Reduced HbA1c and triglycerides IER = CER *p* > 0.05
Varady et al. 2011 [14]	IER = 13% 25% CER = 20% Exercise = 20% Control = 20%	No data	BOCF	IER −5.2 ± 1.1% CER −5.0 ± 1.4% Exercise −5.1 ± 0.9% Control −0.2 ± 0.4%	No data	Increase in LDL particle size IER = CER *p* > 0.05
Harvie et al. 2011 [12]	IER = 20% 25% CER = 13% 9% of potential recruits did not tolerate the 2 day trial of the restricted days of IER and did not enter the study Drop out due to problems adhering to the diet: IER = 5%, CER = 5%. At the end of the trial, 31 of IER (58%) and 46 of CER (85%) subjects planned to continue the diet allocated at randomization.	IER 7 day food diaries at baseline, 1, 3 and 6 months Potential restricted days completed 0–6 months: mean (95% CI) 66 (55%–77%) Overall average daily reduction in energy intake: mean (95% CI) 12 weeks IER −2.40 (−2.94 to −1.87) MJ (~30% restriction) CER −1.65 (−2.11 to −1.18) (~21% restriction), *p* = 0.04 26 weeks IER −2.40 (−2.94 to −1.87) (~30% restriction) CER −1.73 (−2.13 to −1.37) MJ (~21% restriction), *p* = 0.04	LOCF	IER = CER 12 weeks IER −6.3 (4.5)% CER −5.0 (3.6)% *p* = 0.11 24 weeks IER −7.8 (5.9%) CER −6.6 (5.0%) *p* = 0.26	Bioelectrical impedance IER = CER 12 weeks % Body fat loss: IER −2.4 (2.3)% CER −2.0 (2.1)%, *p*= 0.42 Body fat mass: IER −3.8 (2.9) kg CER −3.3 (3.0) kg, *p* = 0.43 24 weeks % Body fat loss: IER −3.4 (3.2%) CER −2.8 (2.7) %, *p* = 0.35 Body fat mass: IER −5.0 (4.4) kg CER −4.4 (3.9) kg, *p* = 0.34 21% of weight lost as FFM in IER and CER *p* = 0.99	Reduction in HOMA insulin resistance IER > CER at 12 and 24 weeks. Mean difference (95% CI) 12 weeks: −17 (−33.2 to −0.2), *p* = 0.046. 24 weeks: −23% (−38 to −8.6)%, *p* = 0.001. Reduced total LDL cholesterol, triglycerides and blood pressure IER = CER *p* > 0.05
Harvie et al. 2013 [13]	IECR = 11% IECR + PF = 26% 25% CER = 33% Drop out due to problems adhering to the diet: IECR 0% IECR + PF 5% CER 5%	7 day food diaries at baseline, 1, 3 and 4 months Potential restricted days 0–4 months mean (95% CI): IECR 76 (67%–81%) IECR + PF 74 (64%–84%) Overall average daily reduction in intake median, %: 12 weeks IECR = −2.97 MJ, −36% IECR + PF = −2.3 MJ, −29% DER = −2.63 MJ, −33% *p* = 0.046 16 weeks IECR = −2.38 MJ, 32% IECR + PF = −2.06 MJ, 26% DER = −2.16 MJ, 25% *p* = 0.765	LOCF	No difference in % weight loss between groups at 12 weeks IECR −6.2 (4.6) IECR + PF −5.7 (3.9) CER −4.3 (4.6). % weight change during 1 month of weight loss maintenance: IECR −0.49 (1.7) IECR + PF −0.34 (0.9) CER −0.13 (0.88) *p* = 0.431 % weight lost as FFM median (95%CI): IECR 36.0 (26.4 to 41.3) IECR + PF 20.4 (13.2 to 27.2) DER 29.3 (25 to 38.1) *p* = 0.048	Bioelectrical impedance reduction in body fat mass at 12 weeks IER > CER, *p* = 0.019 Body fat mass: IECR −3.7 (3.7) kg IECR + PF −3.7 (2.2) kg CER −2.0 (3.3) kg No difference during 1 month of weight loss maintenance: Reduction in body fat/kg IECR −0.58 (1.2) IECR + PF −0.31 (0.7) CER + 0.26 (0.90) *p* = 0.313	Reduction in HOMA insulin resistance IECR > CER. Mean (95% CI) change at 12 weeks: −0.2 (−0.19 to 0.66) unit; *p* = 0.02 After 4 weeks of weight loss maintenance: IECR −0.06 (0.51) IECR + PF + 0.03 (0.6) CER −0.25 (0.53) Unit, *p* = 0.084 Reduced total LDL cholesterol, triglycerides and blood pressure IECR = IECR + PF = CER
Carter et al. 2016 [33]	IECR 16% CER 22%	No data	ITT	IECR −6.2 (3.6)% CER −5.6 (4.4)% *p* = 0.6	DEXA IER 3.8.(2.7)% CER −4.0 (3.2)% *p* = 0.8	HbA1C IER −0.6 (1)% CER −0.5 (0.8)

Mean (SD); IECR + PF, intermittent energy and carbohydrate restriction and ad libitum protein and MUFA; BOCF, baseline observation carried forward; LOCF, last observation carried forward; HOMA, Homeostasis Model Assessment; LDL, low-density lipoprotein; DEXA, dual energy x ray absorptiometry.

**Table 3 behavsci-07-00004-t003:** Summary of research findings and gaps in research comparing intermittent energy restriction/intermittent fasting to CER for weight control and metabolic health.

Outcome	Effects in People Who Are Obese or Overweight	Effects in People Who Are Normal Weights	Effects in Rodent Studies
Weight loss/prevention of weight gain	IER = CER for weight loss in six studies which were not powered to detect differences in weight. The study finding are suggestive but not conclusive of no difference between IER and CER weight [12,13,14,26,27,33]	No long term data	N/A
Proportion of body fat stored as visceral and subcutaneous fat	No data	No data	Mixed results: Reduced visceral and increased subcutaneous fat in female C57BL/6J mice [50]. No change in male C57BL/6J mice [51]. Increased visceral and decreased subcutaneous fat in 4 week old male Sprague Dawley rats [53] and LDL-receptor knockout mice [54].
Fat cell size	No data	No data	Reduced in male C57BL/6J mice [51]
Hepatic fat	No data	Modest increase after a single 24 h fast in men not women [46]	Mixed results: Deposition in IER > CER [53,54] IER = CER [87].
Intra myocellular triglycerides	No data	Modest increase after a single 48 hour fast in women but not men [45]	No data
Insulin sensitivity	Mixed results IER > CER (HOMA) [12,13] Reduced HbA1c IER = CER *p* > 0.05 [27,33]	Mixed results IER > CER [16] IER = CER [17]	Mixed results IER > CER total body and hepatic insulin sensitivity [73]. IER < CER peripheral insulin sensitivity [74] IER > CER hepatic insulin sensitivity [74]
Fat free mass	IER = CER [12,26]	No data	No data
Resting energy expenditure	IER = CER [26,28] IER > CER [18]	No comparison data	No data
Metabolic flexibility	No data	No data	IER > CER [77].

IER, intermittent energy restriction; CER, continuous energy restriction; LDL, low-density lipoprotein; N/A, not applicable.

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
