# Peer review of "Potential Benefits and Harms of Intermittent Energy Restriction and Intermittent Fasting Amongst Obese, Overweight and Normal Weight Subjects—A Narrative Review of Human and Animal Evidence"

_behavsci, 2017, doi:10.3390/bs7010004_

Round 1

Reviewer 1 Report

This review highlights inconclusive findings and numerous gaps in knowledge of potential beneficial or adverse effects of intermittent energy restriction in comparison to isoenergetic continuous energy restriction on metabolic health in obese, overweight and normal weight humans.

Major

1.              The Abstract should clarify what exactly is meant by metabolic flexibility

2.              Add more detail into the abstract, particularly a summary of what the review found, currently only weight and insulin resistance is included

3.              The abstracts mentions “5 small short term…” this suggests that only 5 studies were looked at in this review which is not correct - adding more detail may help.

4.              Please provide more of a rationale as to why it is important study normal weight subjects and how they could benefit from IER?

5.              Sub-headings would be of benefit in this paper. Example, adherence is included in the section “Is IER associated with greater weight control than CER?”  

6.              Line 50 is confusing, authors state that all 3 diets, but only 2 are named, that is the two consecutive days of ER and alternate days of ER (ADER). I assume the third one is alternate day fasting? The authors should make it clear in lines 49-50 that alternate day fasting has also been studies as this is unclear at the moment

7.              The search terms included may not have been broad enough to pick up all relevant studies. Important studies have been missed due to this

8.              Line 102-104, it cannot be said that the only study to show a difference in weight loss was ref (11) as there was no significant difference.

9.              The authors have given details about their study but lack detail for some of the other studies for example, line 103-115. This study (ref 11) did not show a difference in weight as the authors stated but there was a greater loss of body fat. Was body composition measured in any of the other 4 studies that showed no difference in weight loss?

10.          Line 262 was protein provided for both groups?

11.          The paper will be very much improved with a much greater degree of referencing throughout: for example, line 35 provide reference for human and animal studies, provide a ref for sentence beginning line 483, sentence beginning line 46 and paragraph beginning 356.

12.          References need to be formatted correctly, inconsistent throughout the text. For example in line 29 references appear as (1) (2) (3), line 88 they appear as (21;22) (10;11) (12;23;24) (25;26) (27) and line 163 as (10;11).

13.          Line 30, ref (5) should be following “type 2 diabetes”

14.          Line 102 states “Four of the selected studies demonstrate…” but 6 references are included

15.          Table 1, provide details of how energy intake was recorded for each studies included were only those where IER was matched to CER?

Minor

1.              Line 260: Add space between “CER(30%”

2.              Line 260-261: “greater preservation of 260 FFM (20% of weight loss) with a higher protein IER (1.2g protein/kg weight) (p<0.05) (54)“ take out sentence  does not make sense

3.              Line 476: typo concludive

4.              Line 445: should be “recommended”

5.              Check data for sentence beginning Line 361 and provide reference

Author Response

Thank you for this review. I have  responded to the suggestions below.

1.              The Abstract should clarify what exactly is meant by metabolic flexibility

 Not done  this as explantion is quite lengthy.

2.              Add more detail into the abstract, particularly a summary of what the review found, currently only weight and insulin resistance is included

 Have reworded

3.              The abstracts mentions “5 small short term…” this suggests that only 5 studies were looked at in this review which is not correct - adding more detail may help.

 Have revised  numbers

4.              Please provide more of a rationale as to why it is important study normal weight subjects and how they could benefit from IER?

 Added  2 references which describe potential benefits for people of all weights

5.              Sub-headings would be of benefit in this paper. Example, adherence is included in the section “Is IER associated with greater weight control than CER?”  

 Have  added a number of sub headings

6.              Line 50 is confusing, authors state that all 3 diets, but only 2 are named, that is the two consecutive days of ER and alternate days of ER (ADER). I assume the third one is alternate day fasting? The authors should make it clear in lines 49-50 that alternate day fasting has also been studies as this is unclear at the moment

 Have reworded

7.              The search terms included may not have been broad enough to pick up all relevant studies. Important studies have been missed due to this

 Have rerun the search and pisked up 3 new  comaprisons of  IER / IF and  CER

8.              Line 102-104, it cannot be said that the only study to show a difference in weight loss was ref (11) as there was no significant difference.

 Have reworded

9.              The authors have given details about their study but lack detail for some of the other studies for example, line 103-115. This study (ref 11) did not show a difference in weight as the authors stated but there was a greater loss of body fat. Was body composition measured in any of the other 4 studies that showed no difference in weight loss?

 Have reworded

10.          Line 262 was protein provided for both groups?  

 Have reworded  to clarify

11.          The paper will be very much improved with a much greater degree of referencing throughout: for example, line 35 provide reference for human and animal studies, provide a ref for sentence beginning line 483, sentence beginning line 46 and paragraph beginning 356.

 Have  improved this throughout

12.          References need to be formatted correctly, inconsistent throughout the text. For example in line 29 references appear as (1) (2) (3), line 88 they appear as (21;22) (10;11) (12;23;24) (25;26) (27) and line 163 as (10;11).

 Have  corrected this throughout

13.          Line 30, ref (5) should be following “type 2 diabetes”

Have reworded 

14.          Line 102 states “Four of the selected studies demonstrate…” but 6 references are included

 Have reworded 

15.          Table 1, provide details of how energy intake was recorded for each studies included were only those where IER was matched to CER?

  To compare adherence and weight loss success between IER and CER we include only randomised comparisons of IER and CER amongst free living individuals where the prescribed diets had been matched for overall energy intake.

Reviewer 2 Report

This is a very worthwhile, topical and informative review of evidence to date regarding IER and IF.   Given the diversity of evidence this is a valid attempt at compiling and organising the highly variable information.  

Overall I can see the logic in how the paper is structured, however, the authors may consider separating the evidence from animal studies to those from human studies more clearly.  It may be that each section is subdivided in this way. Given the main body of the evidence centres on the metabolic effects of IER and CER it may also be prudent to separate the evidence between weight loss studies, and weight loss independent studies.   To this end the authors may wish to rearrange the structure and rethink the content and naming of sections/subsections.

I would also strongly recommend inclusion of more diagrams/figures to summarise the metabolic effects and the proposed mechanisms of IER/IF, if feasible.  

Some specific comments:

Introduction:

It may be prudent to mention that IER (or specifically IF) does include time restricted feeding, even if this is not a focus of the review. 

Line 49-50.  The Catenacci paper published September 2016 dies use ADF for weight loss if you want to include an example of a recent ADF trial.   (Catenacci et al, Pan Z, Ostendorf D, Brannon S, Gozansky WS, Mattson MP, Martin B, MacLean PS, Melanson EL, Troy Donahoo W. A randomized pilot study comparing zero-calorie alternate-day fasting to daily caloric restriction in adults with obesity. Obesity (Silver Spring). 2016 Sep;24(9):1874-83. 
Line 64. Need a full stop and new sentence between intake and Also, or continue within the same sentence.

Methods. 

The methods section could be clearer as to specific hash terms used.  Also why only one database (i.e Medline was chosen).  Whilst I appreciate this is not a systematic review as such, this would describe the robustness and repeatability of the search better. 

Given the manuscript was submitted in September you may have included the Catenacci paper published September 2016 (see Intro), and the Zuo et al paper from August 2016.  Zuo L, He F, Tinsley GM, Pannell BK, Ward E, Arciero PJ. Comparison of High-Protein, Intermittent Fasting Low-Calorie Diet and Heart Healthy Diet for Vascular Health of the Obese. Front Physiol. 2016 Aug 29;7:350.

Line 95 (regarding reference 21).  The inclusion of reference 21 (Hill et al 1989) is a difficult study to disentangle due to the complexity of the design.  The authors may wish to rethink the inclusion of this if it makes interpretation clearer. 

Line 130 – remove the unnecessary a before “carry over effect”

Line 133- May be useful to state what the average ER was for this CER group and how this differed from the IER. 

Line 144 – Do you mean use of IER for weight maintenance or weight maintenance following weight loss with IER (i.e. when stopping IER and return to free eating)?

Line 149 – 152: I believe that the information % after 6 months of dieting was published in reference 10.  But the 12 months data was the unpublished data. 

Line 169- Not sure this is the correct heading to use for this section.  It is more IER whilst in energy Balance rather than in normal weight individuals.  The studies mentioned (13 and 35) are non-obese and hence do include overweight individuals. 

Line 187 – Not sure Taylor’s paper (ref 36) is the right reference to substantiate this statement. 

Line 191 – specify:  ”There are currently no human data concerning long term or chronic effects…” 

Lines 198.  The extent of FFA flux is likely different between IF and IER.  More pronounce in IF, rather than IER. 

Paragraph beginning line 225.  Not sure the inclusion of these animal studies (including LDLr knockout mice) is adding any value.  You are introducing another aspect that muddies the water and mentioning circadian disturbance only adds further un-answered questions rather than supports understanding.   It may be that if you organise into animal and human that this could be covered better (see general comments), otherwise doesn’t seem to fit here. 

Section on resting energy expenditure (from line 273).  May need updating if including the Catenacci paper from Sept 2016.  As they also measured RMR.

Section from line 290 – if including Catenacci paper then need to add to this section too.

Line 327.  The differing effects on HBA1c in the Williams paper (65), may be due to undertaking VLCD for 5 straight days, rather than IER per se. 

Line 343 – It is important to note that the OGTT performed in this paper (ref 42) was not standardised.  I.e. at baseline it was post a 12 hour fast whereas at follow up it was post 36 hour fast.   Hence it could be an acute effect of the 36 hour fast. 

Line 353 – Delete the word more.  You have mentioned the Euglycaemic hyperinsulinaemic clamp already in this section. 

Line 388 – Need a reference at the end of this statement ideally. 

Tables

Make sure you are consistent in the detail for each reference.  Also make sure you state abbreviations in footnotes or spell in full.  E.g. AL = ad libitum.  

Author Response

Thank you for this review . Responses to the suggestions  are written below.

 This is a very worthwhile, topical and informative review of evidence to date regarding IER and IF.   Given the diversity of evidence this is a valid attempt at compiling and organising the highly variable information.  

Overall I can see the logic in how the paper is structured, however, the authors may consider separating the evidence from animal studies to those from human studies more clearly.  It may be that each section is subdivided in this way. Given the main body of the evidence centres on the metabolic effects of IER and CER it may also be prudent to separate the evidence between weight loss studies, and weight loss independent studies.   To this end the authors may wish to rearrange the structure and rethink the content and naming of sections/subsections.

 Have  added sub headings

I would also strongly recommend inclusion of more diagrams/figures to summarise the metabolic effects and the proposed mechanisms of IER/IF, if feasible.  

Some specific comments:

Introduction:

It may be prudent to mention that IER (or specifically IF) does include time restricted feeding, even if this is not a focus of the review. 

 This is clearly stated in the methods

Line 49-50.  The Catenacci paper published September 2016 dies use ADF for weight loss if you want to include an example of a recent ADF trial.   (Catenacci et al, Pan Z, Ostendorf D, Brannon S, Gozansky WS, Mattson MP, Martin B, MacLean PS, Melanson EL, Troy Donahoo W. A randomized pilot study comparing zero-calorie alternate-day fasting to daily caloric restriction in adults with obesity. Obesity (Silver Spring). 2016 Sep;24(9):1874-83. 
Line 64. Need a full stop and new sentence between intake and Also, or continue within the same sentence.

 Have included this paper in metabolic effects but not  when  comparing  IF to  CER  as  the IF  and CER  regimes are not   prescribed to be isoenergetic

Methods. 

The methods section could be clearer as to specific hash terms used.  Also why only one database (i.e Medline was chosen).  Whilst I appreciate this is not a systematic review as such, this would describe the robustness and repeatability of the search better. 

 Have included search terms in appandix 1

Given the manuscript was submitted in September you may have included the Catenacci paper published September 2016 (see Intro), and the Zuo et al paper from August 2016.  Zuo L, He F, Tinsley GM, Pannell BK, Ward E, Arciero PJ. Comparison of High-Protein, Intermittent Fasting Low-Calorie Diet and Heart Healthy Diet for Vascular Health of the Obese. Front Physiol. 2016 Aug 29;7:350. 

Line 95 (regarding reference 21).  The inclusion of reference 21 (Hill et al 1989) is a difficult study to disentangle due to the complexity of the design.  The authors may wish to rethink the inclusion of this if it makes interpretation clearer. 

 Have retained this study for completeness of the review

Line 130 – remove the unnecessary a before “carry over effect”have reworded

Line 144 – Do you mean use of IER for weight maintenance or weight maintenance following weight loss with IER (i.e. when stopping IER and return to free eating)?

have clarified

Line 149 – 152: I believe that the information % after 6 months of dieting was published in reference 10.  But the 12 months data was the unpublished data. 

have clarified

Line 169- Not sure this is the correct heading to use for this section.  It is more IER whilst in energy Balance rather than in normal weight individuals.  The studies mentioned (13 and 35) are non-obese and hence do include overweight individuals. 

The  heading is correct  as this  section is attemping to  adress this question

Line 187 – Not sure Taylor’s paper (ref 36) is the right reference to substantiate this statement.  This  paper reviews this effect

Line 191 – specify:  ”There are currently no human data concerning long term or chronic effects…”  have reworded

Lines 198.  The extent of FFA flux is likely different between IF and IER.  More pronounce in IF, rather than IER. 

 have added this sentence

Paragraph beginning line 225.  Not sure the inclusion of these animal studies (including LDLr knockout mice) is adding any value.  You are introducing another aspect that muddies the water and mentioning circadian disturbance only adds further un-answered questions rather than supports understanding.   It may be that if you organise into animal and human that this could be covered better (see general comments), otherwise doesn’t seem to fit here. 

 Have included to highlight problems  with studying  certain animal models which may give   effects which do not reflect effects in humans

Section on resting energy expenditure (from line 273).  May need updating if including the Catenacci paper from Sept 2016.  As they also measured RMR.

 Have included

Section from line 290 – if including Catenacci paper then need to add to this section too.

Line 327.  The differing effects on HBA1c in the Williams paper (65), may be due to undertaking VLCD for 5 straight days, rather than IER per se. 

 This is stated

Line 343 – It is important to note that the OGTT performed in this paper (ref 42) was not standardised.  I.e. at baseline it was post a 12 hour fast whereas at follow up it was post 36 hour fast.   Hence it could be an acute effect of the 36 hour fast. 

 Have clarified

Line 353 – Delete the word more.  You have mentioned the Euglycaemic hyperinsulinaemic clamp already in this section. 

 Have reworked

Line 388 – Need a reference at the end of this statement ideally. 

 Have reworked

Tables

Make sure you are consistent in the detail for each reference.  Also make sure you state abbreviations in footnotes or spell in full.  E.g. AL = ad libitum.  

 Have  checked tables

Reviewer 3 Report

Dear authors

Thank you for the opportunity to review your investigation of the potential benefits and harms of IER compared to CER. This is certainly an area of current interest.

I do wonder if the title should reflect the fact that you are making comparisons?

Overall it is an extremely well written submission, although perhaps slightly long. Given how well presented the paper is I have very few minor comments but just a few areas where it would be great to have some clarification. Also some recommendations.

1) Why a narrative review rather than systematic review? You are only including RCTs. If a systematic style review then one would have expected some quality assessment of the papers included and tighter inclusion/exclusion criteria and also a figure showing why certain papers were excluded. I think this would have been beneficial - for example would have been good to know how far back you searched the literature, if the papers were all presented in the English language etc and to have better detailed your population group - I am assuming adults rather than children?

2) I feel studies looking at IF, and discussion of IF generally, should have been excluded and your review to have focussed on IER v. CER and just a sentence rationalising the decision to exclude IF. Similarly just human studies - I don't feel the rodent studies actually provided any additional value and if anything 'muddied the water'.

3) the section 'Is IER associated with greater weight control than CER' is slightly biased towards your own studies in the depth of critiquing included. Could this section be slightly more balanced and critique the other 3 studies with the same level of detail? I am not sure that you need to refer to 10 and 11 as your studies but instead reference them as you would the others. I did find the referencing style slightly difficult to follow eg (10-12;21) (11) - why are they listed in this way?

4) you refer to 'normal' intake on a couple of occasions - is this normal at an individual level?

5) you also refer to 'feed' days - for clarity would it be better to use non IER days?

6)line 97/8 - you refer to completers but with no definition.

7) lines 164/65 you refer to high levels of dietetic support and sometimes meal provision - it would have been good to have more detail about this  in your critiquing of the studies

8) peripheral and hepatic insulin resistance - might the protein:CHO ratio of the IER regime be an important variable? I would have liked to have seen a little more about the protein and CHO content of IER in your conclusions too.

9) is there an optimal IER regimen - you mention optimal duration in your opening sentence but then don't discuss.

10) conclusion - I would have liked to have seen something about the duration and practical implications for example the use of IER after a period of following a VLCD?

11)figure 1 - is this referenced in the text? - to have added value it would be very useful to include a dotted line showing the mean % energy ER actually achieved.

12) Table 1 - I note that exercise was included in some of the intervention arms but not discussed - ? an important variable?

13) Table 3 - would have been good to have seen compliance included as an outcome measure.

Author Response

Thank you for this review. I have  responded to the suggestions below.

Why a narrative review rather than systematic review? You are only including RCTs. If a systematic style review then one would have expected some quality assessment of the papers included and tighter inclusion/exclusion criteria and also a figure showing why certain papers were excluded. I think this would have been beneficial - for example would have been good to know how far back you searched the literature, if the papers were all presented in the English language etc and to have better detailed your population group - I am assuming adults rather than children?

I have included  the medline search as an appendix

I feel studies looking at IF, and discussion of IF generally, should have been excluded and your review to have focussed on IER v. CER and just a sentence rationalising the decision to exclude IF. Similarly just human studies - I don't feel the rodent studies actually provided any additional value and if anything 'muddied the water'.

 Included  all animal and human studies and if studies to  give an overview of the  literature

3) the section 'Is IER associated with greater weight control than CER' is slightly biased towards your own studies in the depth of critiquing included. Could this section be slightly more balanced and critique the other 3 studies with the same level of detail? I am not sure that you need to refer to 10 and 11 as your studies but instead reference them as you would the others. I did find the referencing style slightly difficult to follow eg (10-12;21) (11) - why are they listed in this way?

4) you refer to 'normal' intake on a couple of occasions - is this normal at an individual level?

 Reworded  as  estimated requirements

5) you also refer to 'feed' days - for clarity would it be better to use non IER days?

 Used the term non restricted days

6)line 97/8 - you refer to completers but with no definition.

Have clarified this sentence

7) lines 164/65 you refer to high levels of dietetic support and sometimes meal provision - it would have been good to have more detail about this  in your critiquing of the studies

 Have included  a new column in  table 1

8) peripheral and hepatic insulin resistance - might the protein:CHO ratio of the IER regime be an important variable? I would have liked to have seen a little more about the protein and CHO content of IER in your conclusions too.

 The diets have been described. Detailed discussion of  macronutrient  compsoition and insulin is beyond the scope of this review.

9) is there an optimal IER regimen - you mention optimal duration in your opening sentence but then don't discuss.

This is discussed in section Is there an optimal IER regimen?  On page  15.

10) conclusion - I would have liked to have seen something about the duration and practical implications for example the use of IER after a period of following a VLCD?

Very  scant  data on this topic , except recent   zuo study which compared a 5:2 diet to  standard healthy eating,  so not discussed,  

11)figure 1 - is this referenced in the text? - to have added value it would be very useful to include a dotted line showing the mean % energy ER actually achieved.

 Yes  it is referencd in the text.

12) Table 1 - I note that exercise was included in some of the intervention arms but not discussed - ? an important variable?

Have included  details of  exercise  in table 1 . Exercise was   included in both IER  and CER  groups in a number of studies  and   none of the studies  reported any interactions with  exercise. Added a line to address  this issue  .

Reviewer 4 Report

The review by Harvie and Howell is very timely and would be of great concern in the field of intermittent fasting/energy restriction. Overall, the paper is well-written and the authors attempt to shed light on the current and somewhat confusing literature. There are, however, a few points that need to be clarified.

1. The title of the review implies that the manuscript focuses on human studies. However, there is a great amount of emphasis on animal studies. The use of the word "subjects" when referring to animals could be misleading. Also, the “harms” and “benefits” appear to be short-term.

2. Some clarity is needed in describing the cited studies. For instance, p 11 line 358 describes a study use rats and the same paragraph line 362-363 they refer to a study using mice. Is this the same study?   

3. The take home message is that IER and CER are equal in weight loss. The reason for this, is because there is not a great deal of high-quality studies examining the long-term outcomes. Is this a correct interpretation of the review? 

Author Response

 Thank you for this review.

The title of the review implies that the manuscript focuses on human studies. However, there is a great amount of emphasis on animal studies.

 have  changed title

 The use of the word "subjects" when referring to animals could be misleading. Also, the “harms” and “benefits” appear to be short-term.

  Have   made appropriate changes

2. Some clarity is needed in describing the cited studies. For instance, p 11 line 358 describes a study use rats and the same paragraph line 362-363 they refer to a study using mice. Is this the same study?   

have checked references

3. The take home message is that IER and CER are equal in weight loss. The reason for this, is because there is not a great deal of high-quality studies examining the long-term outcomes. Is this a correct interpretation of the review? 

 have  simplified conclusion

Round 2

Reviewer 1 Report

The authors have done a good job with the manuscript taking on board the suggestions and comments of the reviewer. There are a few typos throughout the text, but I do not have any further comments